# Treatment of Cesarean Scar and Cervical Pregnancies Using the Ovum Aspiration Set for Intrachorial Methotrexate Injection as a Conservative, Fertility-Preserving Procedure

**DOI:** 10.3390/medicina59040761

**Published:** 2023-04-14

**Authors:** Isabell Ge, Carmen Geißler, Alexandra Geffroy, Ingolf Juhasz-Böss, Philipp Wiehle, Jasmin Asberger

**Affiliations:** 1Department of Obstetrics and Gynaecology, University Hospital of Basel, 4056 Basel, Switzerland; 2Breast Center, University Hospital Basel, University of Basel, 4056 Basel, Switzerland; 3Department of Obstetrics and Gynecology, Medical Center—University Hospital Freiburg, 79106 Freiburg, Germany; 4Faculty of Medicine, University of Freiburg, 79106 Freiburg, Germany

**Keywords:** cesarean scar pregnancy, cervical pregnancy, ectopic pregnancy, methotrexate, intrachorial injection, fertility

## Abstract

*Background and Objectives*: Cesarean scar and cervical pregnancies are rare forms of ectopic pregnancies, occurring in 1 in 2000 and 1 in 9000 pregnancies, respectively. Both entities are medically challenging due to their high morbidity and mortality potential. *Materials and Methods*: In this retrospective study, we analyzed all cesarean scar and cervical pregnancies from 2010 to 2019 in the Department of Gynecology and Obstetrics of the University Hospital Freiburg, treated with both intrachorial (using the ovum aspiration set) and systemic methotrexate application. *Results*: We identified seven patients with a cesarean scar and four patients with cervical pregnancies. At diagnosis, the median gestational age was 7 + 1 (range: 5 + 5–9 + 5) weeks and the mean value of ß-hCG was 43,536 (range: 5132–87842) mlU/mL. On average, one dose of intrachorial and two doses of systemic methotrexate were administered per patient. The efficacy rate was 72.7% with three patients (27.3%) needing an additional surgical or interventional procedure. The uterus was preserved in 100% of the patients. Out of the eight patients with follow-up data, five reported subsequent pregnancies (62.5%) that resulted in six live births. None had recurrent cesarean scars or cervical pregnancies. In the subgroup analyses, when comparing cesarean scar pregnancies to cervical pregnancies, patient characteristics, treatment modality, and the outcome did not differ significantly, except for parity (2 versus 0, *p* = 0.02) and the duration since the last pregnancy (3 vs. 0.75 years, *p* = 0.048). When comparing cases with successful and failed methotrexate-only treatments, the maternal age was significantly higher in the successful group (34 vs. 27 years, *p* = 0.02). Localization of the gestation, gestational and maternal age, ß-hCG, and history of preceding pregnancies were non-predictive for the efficacy of the treatment. *Conclusions*: The combined application of intrachorial and systemic methotrexate for the treatment of cesarean scar and cervical pregnancies has been proven effective, well-tolerated, organ- and fertility-conserving with a low complication rate.

## 1. Introduction

Cesarean scar (CSP) and cervical pregnancies (CP) are rare forms of ectopic pregnancies. While CSPs occur in approximately 1 out of 2000 pregnancies, CPs are even less frequent at 1 in 9000 pregnancies, which accounts for <1% of ectopic pregnancies [1,2]. Increasing rates of cesarean delivery elevate the number of pregnant women with a cesarean section (CS) scar worldwide [3,4]. Therefore, it can be assumed that the CSP rate will also increase in the future. Yet CSPs and CPs are medically challenging due to their high morbidity and mortality potential. Facing the possible complication of an unstoppable hemorrhage, invasive procedures such as hysterectomy or uterine artery embolization are frequently required, which leads to a loss of fertility. Therefore, in order to avoid these risks, early diagnosis and appropriate management of both conditions are indispensable.

Presently, there is no uniform classification system for CSPs. However, determining the exact location of the gestational sac is necessary to assess the patient’s risk and to advise whether to terminate or continue the pregnancy [5]. The ESHRE (European Society of Human Reproduction and Embryology) working group on ectopic pregnancy briefly described CSPs in 2020 and published recommendations on the terminology of normally sited and ectopic pregnancies [6]. In 2022, Jordans et al. published a sonographic assessment and reporting system for CSP, the modified Delphi method, to standardize sonographic evaluation in early pregnancy. Depending on the localization of the gestational sac, CSPs can be divided into three groups: crossing the uterine cavity line (CUL), crossing the serosal line (CSL), and not crossing either line but embedded in the myometrium (NUL/NSL). However, the value of the evaluation method on patients’ risk and therapy choice has yet to be determined [7].

For the therapy of CSP and CP, a variety of management strategies exist including conservative and operative procedures [8]. Which management may contribute to better reproductive outcomes remains unknown. As a conservative procedure, methotrexate (MTX), a folic acid antagonist that is highly toxic to rapidly replicating tissues, can be injected intramuscularly (systemic) or locally into the gestation sac, via a transabdominal or transvaginal route under sonographic guidance [9,10]. The transvaginal approach entails the benefit of a shorter distance to the gestational sac with fewer obstacles to avoid. Thus, this access route is associated with a lower risk of bladder injury compared to the transabdominal approach. Recently, a published study evaluated the success of local MTX treatment and its side effects in patients diagnosed with CSP [11]. Cagli et al. indicate that transvaginal ultrasound-guided single-dose local MTX treatment is an effective, safe, and fertility-preserving treatment method for CSP [11]. M. Yamaguchi et al. also described a single local MTX injection as safe and effective for the treatment of CSP [12].

In our institution, the Department of Gynecology and Obstetrics at the University Hospital Freiburg in Germany, the standard of care for treating CSPs and CPs includes the application of systemic and intrachorial MTX via transvaginal approach. In this retrospective study, we aimed to determine the feasibility, complications, and outcome of the procedure as well as factors predicting its efficacy as an interventional, organ- and fertility-preserving method.

## 2. Materials and Methods

### 2.1. Patient Group

We queried our institutional database to identify CSPs and CPs treated from 2010 to 2019 using intrachorial and systemic MTX. The identification was achieved using the German Operation and Procedure Classification System (OPS-) Codes, which were assigned to each patient who underwent the procedure of an intrachorial injection of MTX.

Patient data were extracted from electronic medical records. Basic characteristics such as age, parity, gravidity, previous pregnancies, etc., were obtained from the physician’s letter. Diagnosis, process, and outcome of the pregnancy were documented in ultrasound and outpatient visit reports. Serum ß-human chorionic gonadotropin (hCG) values were obtained from laboratory workups. The application of MTX was described in the surgery and chemotherapy reports. We tried, by telephone, to further assess the follow-up information of patients who were lost during the follow-up or whose outcome was unclear.

### 2.2. Application of MTX

The following dosages were used for the MTX applications: For the intrachorial (IC) injection, 50 mg MTX was administered in 1 to 5 mL of sodium chloride. For the intramuscular (IM) injection, 50 mg MTX/m^2^ body surface was administered.

The intrachorial injection was always conducted in the operating room. Patients either received an analgosedation, general, or regional anesthesia. Using the equipment for a follicular puncture in assisted reproduction technology (ART) cycles, a thin double-lumen 17-gauge needle was inserted transvaginally under ultrasound guidance into the amniotic sac of the ectopic pregnancy. After aspirating off as much amniotic fluid as possible, MTX was injected into the amniotic sac. The patients received the intramuscular MTX injection either on the same day in the ward or during an outpatient visit in our chemotherapy ambulance, up to three days prior to the IC injection. The patients were discharged on the same day or a few days after the procedure.

Prior to the MTX applications, serum laboratory results, including a blood count, electrolytes, urea, creatinine, and ß-hCG were obtained. Blood counts and serum ß-hCG were repeated a median of 4 days after MTX injection. It was decided depending on the ß-hCG decrease whether to repeat IM MTX after one week. If the ß-hCG decrease was adequate and no additional MTX was scheduled, primary care providers could take over further laboratory check-ups for patients who had a long commute to our institution. An example of a CSP at diagnosis before and after treatment with IC/IM MTX is provided in Figure 1.

### 2.3. Statistical Analysis

Statistical analysis was performed using Statistical Analysis System OnDemand for Academics (SAS^®^). We performed t-tests to compare normally distributed mean values, as well as Mann–Whitney U tests for non-normally distributed values. The relationship between categorical variables was assessed using Pearson’s Chi-square test. Logistic regression analysis was used to identify independent variables predicting the binary outcome of treatment with “MTX only” vs. “additional surgical or interventional procedure”.

## 3. Results

### 3.1. Patient Characteristics

We identified 7 CSPs and 4 CPs who underwent IC MTX (Table 1). All ectopic pregnancies were diagnosed in the first trimester (median: 7 + 1 gestational weeks, range: 5 + 5–9 + 5 weeks). Regarding preceding pregnancies, all 4 patients with CP had a previous miscarriage. Patients had high ß-hCG levels with a mean value of 43,536 mlU/mL at diagnosis and 54,510 mlU/mL at therapy initiation. The maximum value at therapy initiation was >150,000 mlU/mL in patient 3, whose therapy was delayed for 3 weeks until 12 + 3 weeks since she initially refused any treatment and wished to continue the pregnancy.

All except one patient with CP received one dose of IC MTX. The patient receiving two doses was an early case from 2011 who did not receive any systemic MTX. The reason for omitting the IM dose is unclear. A median of 2 doses (range: 0–4) of IM MTX was given to the patients. On average, no more MTX was administered when ß-hCG decreased by >90% (range: 77.7–99.9%).

### 3.2. Treatment Outcomes

In total, the efficacy rate of the treatment was 72.7% with 8 out of 11 patients treated successfully with MTX IC +/− IM only. Three out of eleven patients needed an additional surgical or interventional procedure (27.3%). In two patients, dilatation and curettage (D and C) were performed due to bleeding on the first day after IC MTX (patient 5) and the remaining trophoblast material (patient 10). Patient 3 twice received a uterine artery embolization. All three patients were spared a hysterectomy. For the IC injection, patients were hospitalized for a median of two days (range: 0–4 days). MTX was mostly tolerated well, except for patient 1, who reported oral soor, dyspareunia, and mild hair loss, and by patient 8, who reported nausea. However, patient 1 did not report any side effects during the active treatment period until the follow-up telephone interview three months later, thus, missed the opportunity to receive folinic acid. Out of the eight patients with follow-up data, five reported subsequent pregnancies (67.7%), which resulted in six live births. Among these five women, patient 10 was mentioned above as undergoing D and C. All women delivered via cesarean section. Among the other three patients without a subsequent pregnancy, two reported regular menstrual cycles.

### 3.3. Subgroup Analyses

When comparing CSPs and CPs, cases with CPs presented a significantly shorter duration since their last pregnancy (0.75 vs. 2 years, *p* = 0.048) and were mostly nulliparous, in contrast to patients with CSPs, who had a history of two previous births on average (*p* = 0.02). Other than that, there were no significant differences in their characteristics, treatment modalities, and outcomes (Table 2).

We further compared patients who received MTX IM/IC only (success) with patients who needed additional operative and interventional treatment (failure), in order to identify potential predictive factors (Table 3). The analysis revealed that the mean maternal age was significantly lower in the failure group (27.3 vs. 34.1 years, *p* = 0.02), however, it was not a significant risk factor as revealed by logistic regression. Other factors, including gestational age, previous pregnancies and deliveries, ß-hCG, and localization of the CSPs, according to the Delphi reporting method, were also not associated with therapy success (Appendix A).

## 4. Discussion

In reviewing the existing body of literature, there are various treatment strategies for CSPs and CPs, including operative (D and C, hysteroscopy, laparoscopy, and transvaginal resection), interventional (uterine artery embolization and high-intensity focused ultrasound), and conservative management, or the combination of all three [13]. Among the conservative approaches, MTX is the most commonly used [13,14].

MTX is a folic acid analog and an antimetabolite, which is used as a chemotherapeutic agent for various neoplasms, including breast and lung cancers, lymphoma, leukemia, etc. Additionally, MTX can be employed as an immune suppressant for the treatment of autoimmune diseases, such as psoriasis, rheumatoid arthritis, and inflammatory bowel disease. Since it inhibits purine and pyrimidine, and therefore, DNA synthesis, in actively proliferating cells, such as trophoblasts, it has also shown efficacy in the treatment of trophoblastic neoplasms, such as choriocarcinoma, as well as molar and ectopic pregnancies [15].

Since there are different forms of ectopic pregnancies, the administration of MTX may vary depending on the location. Usually, ectopic pregnancies occur outside of the uterus with tubal pregnancies being the most common at 95%, followed by ovarian and abdominal pregnancies, which occur in 3% and 1%, respectively, out of all ectopic pregnancies [16]. In these cases, systemic MTX can be administered intramuscularly, with a standard dose of 50 mg/m^2^ body surface. Less commonly, the ectopic pregnancy may be located inside the uterus, although the gestational sac is implanted either in the cervix (CP) or the myometrium of the scar of a previous cesarean section (CSP), two locations that do not allow proper growth. In these cases, a local MTX injection directly into the gestation sac, via a transabdominal or transvaginal approach, can be considered, either alone or in addition to systemic MTX [17]. Other substances, such as fetal intracardial potassium chloride injection (KCl) [18] or oral mifepristone [19] have also been described as conservative approaches for the treatment of CSPs and CPs in the literature.

In a review, which analyzed over 2000 women with CSPs, systemic MTX alone showed a success rate of 75.2% [14]. Along with D and C only, and needle aspiration in combination with MTX/KCl only, these three methods showed the highest complication rates at up to 21% and a higher number of hysterectomies compared to other methods. In comparison, when using local MTX (20–22-gauge needle) only, the complication rate was lower at 4.1%; however, the success rate was lower as well at 64.9%. When combining local and systemic MTX, the success rate and complication rate were more favorable than systemic or local MTX only, at 76.5% and 2.3%, respectively. However, the case number was low with only 34 cases [14]. In a randomized controlled trial with 104 CSP patients receiving either local or systemic MTX, there was no significant difference in the respective success rate with 69.2% vs. 67.3%, although the time period until ß-hCG and sonographic remission was shorter in the systemic MTX group (42 vs. 56 days, *p* = 0.029; 40 vs. 53 days, *p* = 0.046) [19].

Studies that examined the effect of local MTX alone on both CSPs and CPs were published by Yamaguchi et al. in 2014 and 2017. The local injection was performed using a 21-gauge percutaneous transhepatic cholangiography needle. Although showing highly effective results, repeated local injections were needed in 20–25% of cases (2 out of 8 cases with CSP; 3 out of 15 cases with CP) due to inadequate ß-hCG decline [20,21]. A repeated local MTX injection may entail a second anesthetic procedure, longer hospitalization, and additional psychological stress for the patient. In our study, a second local MTX injection was only performed in one early case when no systemic MTX was administered. When using concomitant IM MTX, a single injection of local MTX was sufficient for all patients. Repeated systemic MTX not requiring hospitalization was administered in our outpatient unit.

In 2022, Yamaguchi published an update with an additional 37 cases of CSPs and showed an overall efficacy rate of 93% using single-dose local MTX [12]. Another recent study with 56 cases of CSPs confirmed the high efficacy rate of local MTX [11]. Similar to our approach, a 16-gauge oocyte collection double-lumen aspiration needle was used to inject 2 mL of 50 mg MTX into the gestational sac. In both studies, IM MTX was administered when ß-hCG did not decrease and a single dose of local MTX was not sufficient, further supporting the practice in our institution of concomitant IC and IM MTX, especially for cases where ß-hCG persisted [11].

Another study by Lu et al. assessing the effect of local treatment only for CSPs used absolute ethanol instead of MTX [22]. Interestingly, the injected volume with a mean value of 8.38 ± 2.65 mL was much higher than other studies using local therapy, possibly to reach the effective concentration of ethanol. Repeated injections were required in 6 out of 26 patients (23.1%). In our institution, especially in recent years, 1 mL of NaCl was sufficient to dissolve 50 mg of absolute MTX. A smaller injection volume may be beneficial in preventing potential complications, such as uterine scar rupture or hemorrhage.

In our study, we showed that using a combined treatment method of intrachorial and intramuscular MTX, all patients could be spared a hysterectomy. Over two-thirds of the patients with follow-up data showed preserved fertility with a subsequent pregnancy. Less than a third of patients required additional operative or interventional treatment. The therapy was mostly well tolerated. Except for two cases, (patients 3 and 5) where the patients suffered blood loss, although neither had severe complications. However, patient 3 should be regarded as an exceptional case since she was highly non-adherent. The CSP treatment initiation with MTX was delayed for three weeks to the second trimester with her ß-hCG levels increasing to >150,000 mlU/mL, which was much higher than in all other patients. Although ß-HCG was negative after treatment with IC and IM MTX, she presented with sudden hypermenorrhea and was diagnosed with non-vascularized remaining trophoblast material and an additional arteriovenous fistula, which had formed from the right uterine artery. However, she repeatedly declined hospital admission, missed several of her control appointments, and rejected diagnostics and treatment. She agreed to treatment only when her hemoglobin level dropped dangerously low to 6.9 g/dL and received a uterine artery embolization, which had to be repeated in another uterine artery branch three weeks later due to repeated hemorrhaging. If the embolization had been performed right the first time in the patient, then the vaginal hemorrhaging and heavy blood loss most certainly could have been avoided. Thus, the case of patient 3 led to the conclusion that the conservative approach with MTX alone may not be sufficient once the ectopic pregnancy reaches a certain maturity and size, similar to the case with tubal pregnancies. This conclusion is supported by a study by Tam et al., which identified higher levels of ß-hCG in CSPs as an independent predictive factor for treatment failure with local MTX injections [23].

Additionally, this is the first study evaluating the localization of CSPs, according to the reporting system of Delphi, as a potential predicting factor for therapy success. In the present study, we found no significant association, which could be due to the low sample size. According to Tam et al., whose study included 30 patients with CSPs, exogenous growth of the gestational sac towards the bladder or abdominal wall was significantly associated with the failure of the local MTX injection [23]. One of two cases in our cohort with a failed MTX treatment showed such a growth pattern, with the largest part of the gestational sac crossing the serosal line. Thus, standardized localization assessment of the gestational sac should be encouraged in routine practice and further studies should be performed in order to help identify potential patterns and develop management recommendations.

Some limitations of the present study need to be acknowledged. Apart from the retrospective design, the number of patients was small so the results need to be verified by future studies with more patients.

## 5. Conclusions

In conclusion, treatment strategies for ectopic pregnancies are heterogeneous. Our institutional approach of combining systemic and intrachorial MTX via transvaginal approach has been proven effective, well-tolerated, and organ- and fertility-conserving with a low complication rate. However, in order to evaluate whether this method is superior to other treatment methods, a prospective, randomized controlled trial with more patient cases is needed.

Lastly, as CSPs and CPs remain highly challenging pathologies, their treatment should always be performed in selected hospitals where an interplay of the key disciplines: sonographic diagnostics, chemotherapy, reproductive medicine, interventional radiology, and surgery are ensured, in order to provide the best possible treatment process and outcome.

## Figures and Tables

**Figure 1 medicina-59-00761-f001:**
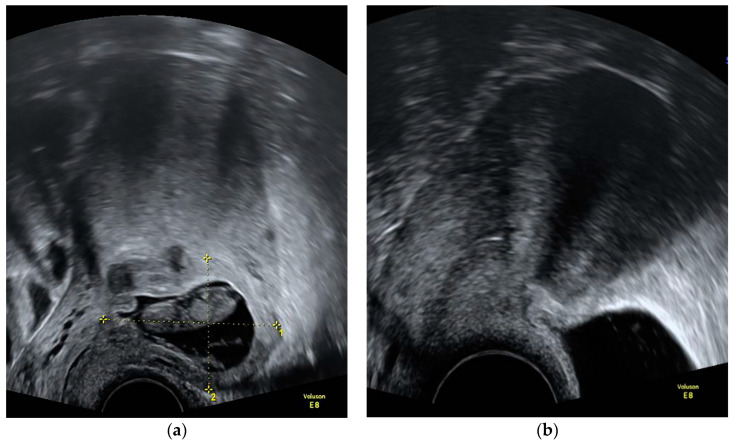
Ultrasound image of a cesarean scar pregnancy before and after treatment with intrachorial and intramuscular methotrexate: (**a**) at initial diagnosis (11 + 1 gestational weeks); (**b**) 4 months after treatment with methotrexate.

**Table 1 medicina-59-00761-t001:** All patients with ectopic pregnancies received IC MTX at the Department of Gynecology and Obstetrics, University Hospital Freiburg from 2010–2019.

Patient No.	Age	GA at Diagnosis/Therapy Start	Previous Pregnancies (Most Recent × Years Ago)	Localization (Modified Delphi Method)	Cycles of MTX IC/IM	ß-hCG at Diagnosis/Therapy Start (mlU/mL)	ß-hCG Decrease When MTX Was Stopped	ß-hCG at Follow-Up (mlU/mL)	H-Days	Outcome
**Cesarean Scar Pregnancy (CSP)**										
1	27	9 + 4 10 + 0	1 × CS (2)	CSL	1/2	38,820/ 34,055	>99.9%	Neg.	1	Regular cycles
2	34	6	3 × CS (3)	NUL/NSL	1/4	29,066	>99.9%	Neg.	1	Missed appointment for follow-up
3	32	9 + 5 12 + 3	2 × CS 1 × AB (3)	CSL	1/2	24,506/ 151,634	91.1%	Neg.	1	AV fistula after 3 m, 2× UAE
4	35	6 + 1 6 + 2	1 × CS (2)	CSL	1/2	24,271	87.7%	168.2	3	Moved abroad
5	28	6 + 0 6 + 5	2 × VD 2 × CS 2 × MC	CSL	1/1	34,865/ 35,172	89.1%	21.1	4	D and C— Bleeding 1st day after MTX
6	37	7 + 1 8 + 0	2 × CS (4)	NUL/NSL	1/0	55,345/ 64,753	77.7%	6.7	3	Pregnancy—primary CS after 2 y
7	35	8 + 4 9 + 2	1 × CS (4)	CUL	1/1	87,842/ 65,457	78.4%	174.9	0	Pregnancy—primary CS after 4 y
**Cervical Pregnancy (CP)**										
8	33	6 + 0 6 + 3	1 × EUP (1) 1 × MC	n/a	1/2	40,661	N/A	Neg.	1	Regular cycles
9	37	7 + 3 7 + 4	1 × MC (0)	n/a	1/2	70,536/ 77,772	92.1%	Neg.	2	Pregnancy— 2 × CS on maternal request after 3 y
10	22	6 + 1 6 + 2	1 × MC (0)	n/a	1/2	67,849	99.9%	Neg.	4	Hysteroscopy + D and C due to rest material Pregnancy— secondary CS after 1 y
11	35	5 + 5 6 + 2	1 × CS (2) 1 × MC	n/a	2/0	5132 8917	94.3%	2.7	2	Pregnancy— primary CS after 1 y

(Abbreviations: MTX: methotrexate; IC: intrachorial; IM: intramuscular; GA: gestational age; H-days: hospitalization days; CS: cesarean section; AB: abortion; VD: vaginal delivery; MC: miscarriage; AV: arteriovenous; m: months; UAE: uterine artery embolization; D and C: dilatation and curettage; y: year(s)).

**Table 2 medicina-59-00761-t002:** Comparison between cesarean scar pregnancies and cervical pregnancies.

Variable	CSP Mean (Range) N = 7	CP Mean (Range) N = 4	*p*-Value
Maternal age (years)	32.6 (27—37)	31.8 (22—37)	0.79
Gravidity	3.4 (2–7)	2.5 (2–3)	0.48
Parity	2 (1–4)	0.25 (0–1)	0.02
Duration since last pregnancy (years)	3 (2–4)	0.75 (0–2)	0.048
Gestational age at diagnosis (weeks)	7 (6–9)	6 (5–7)	0.16
Diagnosis—therapy start (days)	6.5 (1–19)	2.25 (1–4)	0.13
Cycles of IM MTX	1.7 (0–4)	1.5 (0–2)	1.00
ß-hCG (at diagnosis)	42,102 (24,271–87,842)	46,045 (5132–70,538)	0.82
ß-hCG (maximum)	61,697 (24,271–151,953)	60,208 (8917–93,521)	0.96
Pregnancy rate	1/4 (25%)	3/4 (75%)	0.16
Operation/embolization rate	2/7 (28.6%)	1/4 (25%)	0.90
Side effect rate	1/7 (14.3%)	1/4 (25%)	0.66
Duration of hospitalization (days)	1.9 (0–4)	2.25 (1–4)	0.66

(Abbreviations: CSP: cesarean scar pregnancy; CP: cervical pregnancy; HCG: human chorionic gonadotropin; IM: intramuscular; MTX: methotrexate).

**Table 3 medicina-59-00761-t003:** Characteristics of patients with successful and failed MTX (IC/IM) injection.

Variable	Success * Mean (Range) N = 8	Failure ** Mean (Range) N = 3	*p*-Value
Maternal age (years)	34.1 (27–37)	27.3 (22–32)	0.02
Gravidity	3.4 (2–7)	2.5 (2–3)	0.30
Parity	2 (1–4)	0.25 (0–1)	0.54
Duration since last pregnancy (years)	2.25 (0–4)	1.5 (0–3)	0.70
Gestational age at diagnosis (weeks)	7 (5–9)	7 (6–9)	1.00
Diagnosis—therapy start (days)	3.3 (1–6)	8.3 (1–19)	0.58
Cycles of IM MTX	1.6 (0–4)	1.7 (1–2)	0.91
ß-hCG (at diagnosis)	43,959 (5132–87,842)	42,407 (24,506–67,849)	0.93
ß-hCG (at therapy initiation)	43,119 (8917–35,172)	84,885 (35,172–151,634)	0.11

* Treatment with MTX (IC +/− IM) only, ** Need of additional surgical or interventional procedure. (Abbreviations: HCG: human chorionic gonadotropin; IC: intrachorial; IM: intramuscular; MTX: methotrexate).

## Data Availability

The data presented in this study are available on request from the corresponding author. The data are not publicly available due to privacy.

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
