# Peer review of "Treatment of Cesarean Scar and Cervical Pregnancies Using the Ovum Aspiration Set for Intrachorial Methotrexate Injection as a Conservative, Fertility-Preserving Procedure"

_medicina, 2023, doi:10.3390/medicina59040761_

Round 1

Reviewer 1 Report

This manuscript examined whether local and systemic methotrexate administration can improve outcomes (i.e., reduce complications and preserve fertility) in women with cesarean scar and cervical pregnancies. The authors found that combined methotrexate (i.e., intrachorial and intramuscular) delivery in ectopic pregnancies is effective, well-tolerated, organ- and fertility-conserving with a low complication rate. Overall, the manuscript is interesting and well written. I have a few brief comments below:

Abstract:

Please remove word contractions from the manuscript (i.e., didn’t).

Please highlight the abbreviation for methotrexate (MTX) on its first use.

Please include a conclusion statement on what the study has shown.

Introduction:

Please amend line 5 to read ‘… CPS rate will also increase in THE future’.

Paragraph 3, Line 6 – As MTX has been defined in the previous sentence it is no longer required to redefine the abbreviation and MTX can be used for methotrexate throughout the remaining manuscript.

Methods:

Figure 1 legend should be under the figure, not above. Please correct.

Results:

Table 1 – there are some instances where a comma (,) is used instead of a decimal (.), please correct.

Author Response

Dear reviewer,

Thank you for the opportunity to revise our manuscript entitled “Treatment of cesarean scar and cervical pregnancies using the ovum aspiration set for intrachorial methotrexate injection as a conservative, fertility-preserving procedure”.

We appreciate the careful review and constructive suggestions. It is our belief that the manuscript is substantially improved after making the suggested edits. Our point-by-point answers to the comments and recommendations are given below in green font.

In the manuscript, the modified sentences are marked using the "track changes" function.

-------------------------------

This manuscript examined whether local and systemic methotrexate administration can improve outcomes (i.e., reduce complications and preserve fertility) in women with cesarean scar and cervical pregnancies. The authors found that combined methotrexate (i.e., intrachorial and intramuscular) delivery in ectopic pregnancies is effective, well-tolerated, organ- and fertility-conserving with a low complication rate. Overall, the manuscript is interesting and well written. I have a few brief comments below:

Abstract:

Please remove word contractions from the manuscript (i.e., didn’t).

Please highlight the abbreviation for methotrexate (MTX) on its first use.

Please include a conclusion statement on what the study has shown.

Thank you for your comments. We have changed "didn’t" to "did not", spelled out all abbreviations in the abstract and added the following conclusion statement: “The combined application of intrachorial and systemic methotrexate for the treatment of cesarean scar and cervical pregnancies has been proven effective, well-tolerated, organ- and fertility-conserving with a low complication rate.“

-------------------------------

Introduction:

Please amend line 5 to read ‘… CPS rate will also increase in THE future’.

Thank you for your remark. We added “the” to the sentence.

Paragraph 3, Line 6 – As MTX has been defined in the previous sentence it is no longer required to redefine the abbreviation and MTX can be used for methotrexate throughout the remaining manuscript.

Thank you for your suggestion. We highlighted the abbreviation "MTX" at its first use and changed all the following “methotrexate” to “MTX” throughout the manuscript.

-------------------------------

Methods:

Figure 1 legend should be under the figure, not above. Please correct.

Thank you. We moved the legend under the figure.

-------------------------------

Results:

Table 1 – there are some instances where a comma (,) is used instead of a decimal (.), please correct.

Thank you for noticing this. We standardized all numbers by changing commas to decimal.

Thank you very much again for your careful review of our manuscript and your recommendation for publishing. It was an honor to have you as the reviewer, since the manuscript could now be refined in a manner that elevates its quality and meaning for the reader. For that, we are grateful.

Best,

Isabell Ge

Reviewer 2 Report

Dear author, I've evaluated your article titled ‘Treatment of cesarean scar and cervical pregnancies using the ovum aspiration set for intrachorial methotrexate injection as a conservative, fertility-preserving procedure’. I will make some suggestions before its publication.

In the text, it was stated that ‘Blood count and serum ß-hCG were repeated a median of 5 days after MTX injection’. The adoption of this method instead of ß-hCG monitoring on the 4th and 7th days after MTX application can be discussed in the discussion section.

Did researchers use folinic acid for patient 2?

Author Response

Dear reviewer,

Thank you for the opportunity to revise our manuscript entitled “Treatment of cesarean scar and cervical pregnancies using the ovum aspiration set for intrachorial methotrexate injection as a conservative, fertility-preserving procedure”.

We appreciate the careful review and constructive suggestions. It is our belief that the manuscript is substantially improved after making the suggested edits. Our point-by-point answers to the comments and recommendations are given below.

In the manuscript, the modified sentences are marked with the "track changes" function.

-------------------------------

Dear author, I've evaluated your article titled ‘Treatment of cesarean scar and cervical pregnancies using the ovum aspiration set for intrachorial methotrexate injection as a conservative, fertility-preserving procedure’. I will make some suggestions before its publication.

In the text, it was stated that ‘Blood count and serum ß-hCG were repeated a median of 5 days after MTX injection’. The adoption of this method instead of ß-hCG monitoring on the 4th and 7th days after MTX application can be discussed in the discussion section.

Thank you for your remark. After rechecking our patient files, we realized that in our institution the day when the patients receive MTX is labeled as day 1 instead of day 0 (which is the standard). When we regard the MTX treatment as day 0, the ß-hCG monitoring occurred on day 4 (instead of day 5 in the first manuscript) which is in line with the standard monitoring recommendation. We have changed our manuscript accordingly. The monitoring of day 7 was often conducted outside of our institution by primary care doctors. This information has also been added to the revised manuscript for better understanding.

-------------------------------

Did researchers use folinic acid for patient 2?

Thank you for your excellent question. Folinic acid was not used for patient 2 since she tolerated the therapy well. It was also not used for patient 1 who experienced oral soor, mild hair loss etc. This is due to the fact that the patient did not report the extent of her side effects during the active treatment and monitoring period, but only recalled these three months later at our telephone interview, thus missing the opportunity to receive folinic acid. We have added this information to the manuscript.

Thank you very much again for your careful review of our manuscript and your recommendation for publishing. It was an honor to have you as the reviewer, since the manuscript could now be refined in a manner that elevates its quality and meaning for the reader. For that, we are grateful.

Best,

Isabell Ge